# The Role of the Non-Playing Hand during Topspin Forehand in Table Tennis

Sławomir Winiarski [1], Ivan Malagoli Lanzoni [2] and Ziemowit Bańkosz [1,*]

1 Division of Biomechanics, University School of Physical Education in Wrocław, 51-612 Wrocław, Poland; slawomir.winiarski@awf.wroc.pl
2 Department of Biomedical and Neuromotor Sciences, University of Bologna, 40126 Bologna, Italy; ivan.malagoli@unibo.it
* Correspondence: ziemowit.bankosz@awf.wroc.pl

**Abstract:** Theoretical tutorials and the scientific literature do not provide information on the proper use of the non-playing hand in table tennis. This study aimed to evaluate the course of the movement in the joints of the non-playing limb during a table tennis topspin forehand stroke (played after a backspin ball) and to determine the inter-individual movement variability. The study involved 12 male table tennis players (178.7 ± 5.5 cm, 70.0 ± 6.6 kg, 23 ± 3 y) at a competitive level. The participants performed one topspin forehand as a response to a backspin ball. Kinematics were measured using an Inertial Motion Unit–MR3 myoMuscle Master Edition system. Changes in the angles of the upper limb joints (with particular emphasis on the non-playing hand) during the forehand topspin were analyzed. A novel method of normalized function of variance was used to characterize areas of high/low variability of movement. Most of the movements in the joints of the non-playing limb were performed symmetrically to the playing one, especially in the hitting phase. A rapid change of direction characterizes these movements, just before or during the hitting phase, which may indicate a supportive, 'driving' character for these movements. High inter-individual variability for the duration of the entire movement cycle in both limbs was observed; higher in the non-playing limb. This perhaps indicates a greater degree of individualization on the non-playing side.

**Keywords:** kinematics; table tennis; non-playing limb; movement variability

## 1. Introduction

Technique in table tennis refers to the motor activities related to hitting the ball with a racket in an appropriate way, which can be achieved through proper footwork. Most of these actions are hitting movements, from the group of 'batting tasks' [1], performed with the whole body, and using the principle of sequential movements, known as the proximal to distal sequences [2,3]. Consequently, individual body segments move in different phases in a variety of ways, in a coordinated kinematic chain. It has been observed that the majority of these movements are 'pre-stretch' or 'countermovement' actions, known as the stretch–shortening cycle [4,5], which increase the performance of the muscles involved in these movements. The principles of using the kinematic chain in sports technique have previously been the subject of many studies and widely described [6]. The kinetic chain refers to the linking of multiple segments of the body, which allows for the transfer of forces and motion [7]. In many sports, the lower limbs and trunk are the base, generating energy that is ultimately transmitted through the throwing (or bouncing) arm and hand, resulting in the throwing (or hitting) of the ball. Some authors speak of two strategies for using the kinematic chain in sport. For example, in tennis, Eliott [8] states that, whenever a player is trying to generate hitting power, he must coordinate the movement in such a way as to obtain the highest racket speed at the moment of hitting the ball (using the stretch–shortening cycle and proximal-to-distal sequences). On the other hand, when the

precision of the stroke is dominant, the player must reduce the force. In this case, fewer body segments are involved in the movement and they act as a stabilizing unit. Any dysfunction or misuse of a particular body segment can have a negative impact on the effectiveness of the kinematic chain. This can also increase the risk of injury [6].

The importance of individual body segments in complex hitting movements in table tennis has already been partially reported in the literature. Iino, Mori, and Kojima [9] studied the influence of movements in the joints of the playing limb on racket velocity during topspin backhand strokes. They noted the importance of wrist dorsiflexion and elbow extension movements in the studied strokes. These authors also found that the importance of these movements in relation to racket speed was associated with the difference in upper limb configuration. Iino and Kojima [10,11] evaluated and determined the importance of internal rotation of the shoulder joint during a topspin forehand stroke. They also noted the energy transfer from the trunk rotation to the playing limb, to generate more force when the racket contacts the ball. Malagoli Lanzoni et al. [12] evaluated the kinematic characteristics of topspin forehand strokes, finding differences in the function of the different body segments, depending on the direction of impact (ball location on the table). Other authors studied the role of the lower limbs during topspin strokes. Marsan et al. [13] pointed out the importance of, and differences in, energy generation in the hip joints during offensive backhand and forehand strokes. The role of trunk rotation (around the vertical axis) for increasing racket velocity during topspin strokes was also noted in an earlier work by Bańkosz and Winiarski [3,14]. In the works available in the literature on asymmetrical sports (use of one hand to throw or hit the ball), little attention has been paid to describing the non-playing (or non-dominant) limb movements. A few works report the importance of the playing–nonplaying hand complex during the execution of a serve in tennis [15]. Theoretical tutorials and the scientific literature do not provide information on the proper use of the non-playing hand. Owing to the importance of the rotational movement of the trunk and the possibility of a subsequent increase in racket speed during the execution of a topspin stroke, it seems advisable to create the possibility of increasing the momentum in the movement around the vertical axis of the body. Similarly to when performing a pirouette in figure skating, coordinating a hitting motion in table tennis could involve bringing the limbs closer to the torso (decreasing the moment of inertia) to increase the angular velocity around the long axis of the body (conservation of angular momentum). The movement of the non-playing limb may also affect the coordination of the whole movement; i.e., the stabilization of the position, the coordination of the whole body (or torso), and the movement of the playing limb. Therefore, it seems to be an important problem whether high-level players in table tennis use the non-playing hand in this way.

Additionally, it seems essential to investigate how large are the inter-individual variation in the movements of the non-playing limb in table tennis players, as this may give information about variations in their training. Therefore, this study aimed to evaluate the course of movement in the joints of the non-playing limb during a topspin forehand (played after the backspin ball), during the stroke cycle, and to determine the inter-individual variation in this area. The results of such a study, besides the cognitive aspects, can provide instruction and vital information to coaches and table tennis players.

## 2. Materials and Methods

### 2.1. Participants

The study involved 12 male table tennis players. All the players presented a competitive level and played in the Polish Superliga. All of the participants declared having more than 12 years of experience in table tennis and presented an offensive style of game. Ten players were right-handed and two were left-handed. The players' dominant hand was established according to which hand they used for playing [16]. Average body height was $178.7 \pm 5.5$ cm, whereas body weight was $70.0 \pm 6.6$ kg. The average age of the group was $23 \pm 3$ y.

All the participants signed an informed consent before the research and were informed about the purpose of the study. Exclusion criteria for the study participants were pain or recent injury. All procedures performed in this study received positive approval from the Senate's Research Bioethics Commission at the University School of Physical Education in Wrocław, Poland (Ethics IRB number 34/2019).

## 2.2. Laboratory Set-Up

Kinematics were measured using a MR3 myoMuscle Master Edition system (myoMotion™, Noraxon, Scottsdale, AZ, USA). The myoMotion system consists of a set of (1 to 16) sensors using inertial sensor technology. The MyoMotion uses advanced, medical-grade inertial motion units (IMU) that measure 3D change in angular position with an accuracy level within 1–2 degrees of a camera-based systems, and with an accuracy comparable to optoelectronic systems [17,18]. Based on the so-called fusion algorithms, the information from a 3D accelerometer, gyroscope, and magnetometer was used to measure the 3D rotation angles of each sensor in absolute space (yaw–pitch–roll; also called orientation or navigation angles) [19]. The sensors were located on the body in the study, according to the myoMotion manual.

Elastic straps and self-adhesive tape were used to attach the sensors to the participant's body. The sensors were placed bilaterally, so that the positive x-coordinate on the sensor label corresponded to a superior orientation for the trunk, head, and pelvis. For the limb segment sensors, the positive x-coordinate corresponded to a proximal orientation. For the foot sensor, the x-coordinate was directed distally (to the toes). Before the measurement in each trial, all the participants were checked and the system was calibrated according to the manufacturer's recommendations. The recording speed of the piezoelectric sensor was 100 Hz per sensor for the whole 16-sensor set. Noraxon's IMU technology mathematically combines and filters incoming source signals on the sensor level and transmits the 4 quaternions of each sensor. Build fusion algorithms and Kalman filtering (digital bandpass finite impulse response filter (FIR)) were used in the study. This mode allowed direct access to all unprocessed, raw IMU sensor data.

## 2.3. Experimental Procedures

The participants performed one task of topspin forehand as a response to a backspin ball, repeated 15 times. Each player was asked to hit the ball at the 'highest point' stage of its flight and to reach the marked area in the corner of the table (30 × 30 cm) diagonally (after the instruction: 'Play diagonally, accurately, and as hard as you can') [5,18,19]. After video analysis only successful shots considered 'on table' and played diagonally were recorded for further calculations (balls hit out of bounds, missed balls, and balls hit into the net were not considered). The balls were played by a table tennis robot (Nevgy Robo Pong Robot 2050, Nevgy Industries, Hendersonville, TN, USA, Figure 1) with constant parameters of rotation, speed, direction, and flight trajectory. The settings of the robot were as follows: rotation type = backspin; speed and spin (where 0 is the minimum, and 30 is the maximum) = 11; right position (rightmost position to which the ball is delivered) = 4; wing (robot's head angle indicator) = 9.5; frequency (time interval between balls thrown) = 1.4 s.

Each player had 3 to 5 familiarization trials before the task. In order to avoid any influence of materials on the examined kinematic parameters, the same racket was used with the following parameters: blade = Jonyer-H-AN (Butterfly, Japan), rubbers = Tenergy 05 (Butterfly, Japan), and thickness of the sponge = 2.1 mm. The experiment was carried out with plastic Andro Speedball balls, 3S 40+ (Andro, Germany) on a Donic Persson 25 table (Donic, Germany).

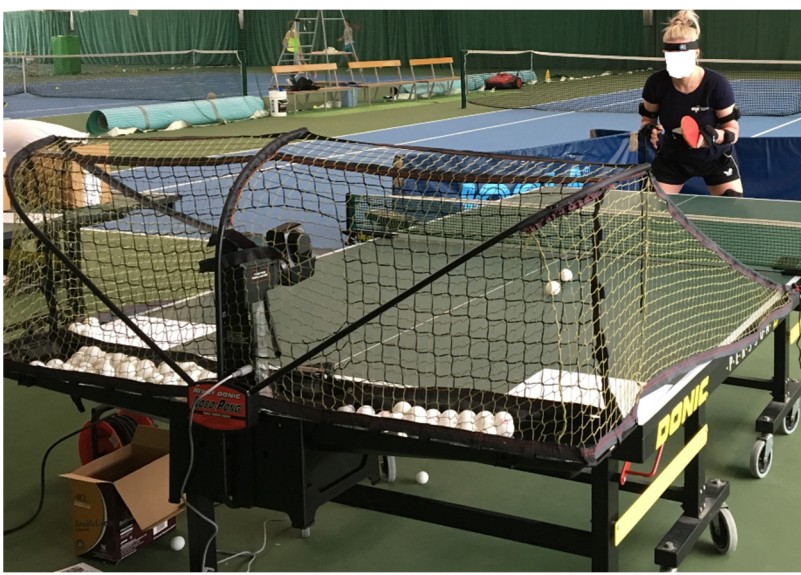

**Figure 1.** Research station.

*2.4. Kinematic Analysis and Calculations*

A simplified biomechanical model was adopted, based on ISB recommendations and the predominant plane of movement, as described by Kontaxis et al. [20]. Based on the adopted sequence of Euler angles and following our previous study [5,18,19], the following angles were computed: (1) wrist radial abduction–adduction: movement of wrist relative to the radius and measured between the upper arm and hand sensors; adduction (or ulnar deviation) is negative while abduction (or radial deviation) is positive; (2) wrist supination–pronation: movement of wrist relative to the radius along the axis and measured between the upper arm and hand sensors; pronation is a positive rotation, supination is a negative rotation; (3) wrist flexion–extension: movement of the wrist relative to the radius along the transversal axis and measured between upper arm and hand sensors; a negative sign denotes extension, while positive is flexion; (4) elbow flexion–extension: movement of the forearm relative to the humerus along the transversal axis; a negative sign denotes (hyper) extension, while positive is flexion; (5) shoulder internal–external rotation: movement of the humerus relative to the thorax in the transversal plane; a negative sign denotes internal (medial), while positive is external (lateral) rotation; (6) shoulder abduction–adduction: movement of the humerus relative to the thorax in the frontal plane; negative sign denotes adduction, while positive is abduction; (7) shoulder flexion–extension: movement of the humerus relative to the thorax in the sagittal plane; a negative sign denotes extension, while positive is flexion [5,18,19].

To describe and assess the specific events of the cycle, the movement of the playing hand was observed: (1) the ready position is starting position, where the hand is not moving after the previous stroke, and just before the swing back; (2) backswing, which is the moment when the hand stops and changes direction from backward to forward in the sagittal plane after the swing; (3) AccMax, which is the moment when the playing hand reaches maximum acceleration during the forward movement; (4) forward, when the hand stops and changes the direction from forward to backward in the sagittal plane, after the stroke. After the moment of forward, the player moves the hand back to the ready position and next cycle starts. The phases between defined events are as follows: (1) back to ready position phase (between the forward and ready positions); (2) backswing phase (between ready position and backswing); (3) hitting phase (between the backswing and Accmax); and (4) forward end phase (between AccMax and forward) [5,17,18].

For each angle of investigation, a population mean ± standard deviation (SD) was depicted. Normalized function of variance (NFV) was used to characterize areas of high/low variability (low/high repeatability) of movements, and the information was transferred to

a horizontal bar representation (Figure 2). The regular CV is the standard deviation (SD) divided by the population mean. To overcome the problem of CV overestimation for small angles, for each angle waveform, the mean was shifted by 1SD in the positive direction and followed the resulting formula:

$$\mathrm{NFV}(\%) = \frac{\mathrm{SD}}{\mathrm{Mean} + \mathrm{SD}} \cdot 100\% \tag{1}$$

The adopted convention was that:

- NFV < 20% was interpreted as low variability (dark blue color on the horizontal bar)
- 20% < NFV < 40% = average variability (blue color)
- 40% < NFV < 100% = high variability (light blue color)
- 100% < NFV < 150% = very high variability (very light blue)
- NFV > 150% = extremely high variability (white color)

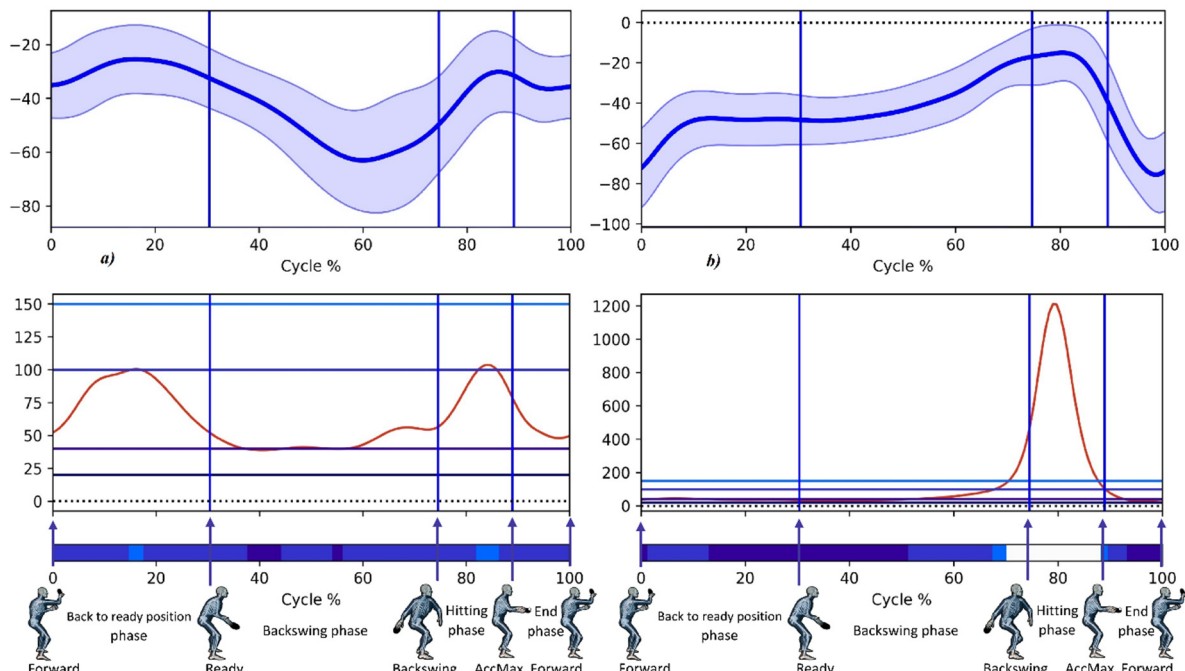

**Figure 2.** Methodology of graph creation. Left and right shoulder rotation (above graph) as an example of low (**a**) vs. high (**b**) variability results. The variance (bottom graph) information exceeding predefined 20, 40, 100, or 150% thresholds (horizontal blue lines) was transferred to the color bar representation. Vertical blue lines indicate the forward, ready, backswing, and AccMax positions in the movement cycle.

## 3. Results

By evaluating the angular waveforms in the various joints and planes of the non-playing limb, the changes in angles during the movement cycle in this limb were evaluated and compared with those occurring in the playing limb. Next, the degree of interindividual variation in the course of the movement in both upper limbs was assessed using the normalized function of variation (NFV).

Shoulder flexion–extension: The shoulder joint of the non-dominant limb exhibits an extension in a range from an average of 40 degrees to about 0 degrees during the back to ready position (Figure 3). Then, before the swing with the dominant hand begins, a flexion is initiated, which lasts until about 3/4 through the backswing phase, to an average of 30 degrees. Then shoulder extension is again observed to about 0 degrees. During the hitting phase, flexion occurs, already in a slightly larger range, up to about 40 degrees (but the SD values are large) at the end of this phase. During the hitting phase the movement in the shoulder joint of the playing limb has a larger range (ca. 0–100 degrees), first it is an extension, then from the end of the backswing phase to the end of the hitting phase it is a

flexion (Figure 3). The movement of the non-playing hand varies individually (as far as the angle values are concerned), but this was quite similar for all subjects. The variability between the players in the non-playing limb was high or very high throughout the cycle. In the playing limb this variation is smallest in the striking phase and in the ready phase, as evidenced by the medium and small NFV values (Figure 3).

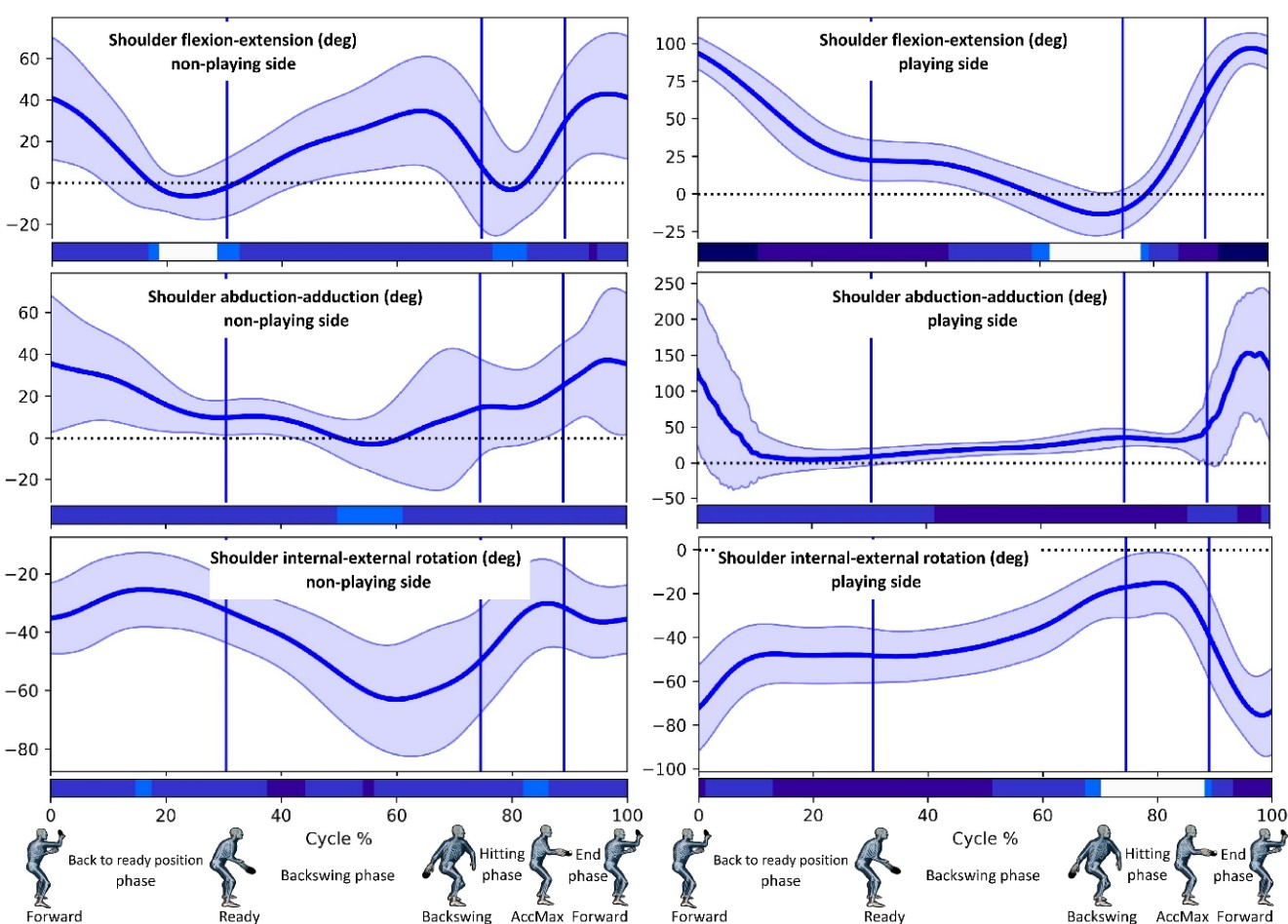

**Figure 3.** Shoulder kinematics for the non-playing (on the left) and playing side (on the right). The movement of the shoulder was evaluated separately for the sagittal plane (flexion–extension), frontal plane (abduction–adduction), and transversal plane (internal–external rotation). The information about variance in the movement exceeding predefined 20, 40, 100, or 150% thresholds was transferred into color bar representation and placed under each graph. Vertical blue lines indicate the forward, ready, backswing, and AccMax positions in the movement cycle.

Shoulder abduction–adduction: The movement at the shoulder joint in the non-playing limb in the frontal plane is an inter-individual varying movement in the hitting phase (large SD values and NFV score, Figure 3). Throughout both the back to ready position and backswing phases there is a slight movement of adduction and maintenance of the limb at about 0 degrees, thus in a medium position. In the last part of the backswing phase abduction begins, up to about 40 degrees at the end of the hitting phase. This movement is accompanied by a large SD of the angles achieved. This movement has similar characteristics in the playing limb, but the ranges of motion are much greater. In the playing limb, the movement range in the hitting phase is about 120 degrees, with less interindividual variation (small and medium NFV) in the backswing and hitting phases than in the non-playing limb.

Shoulder internal–external rotation: The players keep the non-playing limb in the shoulder joint in an internal rotation during the entire stroke cycle. During the back to ready position phase and most of the backswing phase, this rotation increases to about

60 degrees (with a large SD, Figure 3). At the end of the backswing phase, the direction of movement is changed to external rotation. This movement takes place until the moment of maximum acceleration of the playing hand; to about 35 degrees. Then the hand remains in internal rotation, at about 30 degrees, until the beginning of the back to ready position phase. An extensive range of internal rotations in the playing limb was observed during the hitting phase; from about 20 to 80 degrees. The NFV test showed large and very large inter-individual variability in the vast majority of the movements in the non-playing limb. This variation was smaller in the non-playing limb at the end of the impact phase and during the back to ready position phase.

Elbow flexion–extension. The angle during the stroke cycle at the elbow joint on the non-playing side remained in flexion for the entire back to ready position and part of the backswing (in flexion about 80 degrees, Figure 4). In the middle of the backswing phase, the player extends the elbow until the beginning of the hitting phase; to about 30 degrees of flexion. He then flexes the limb throughout the hitting phase. The movement at this joint on the playing side has very similar characteristics. In both limbs, small and medium NFV values indicate little variation in movement (moderate repeatability).

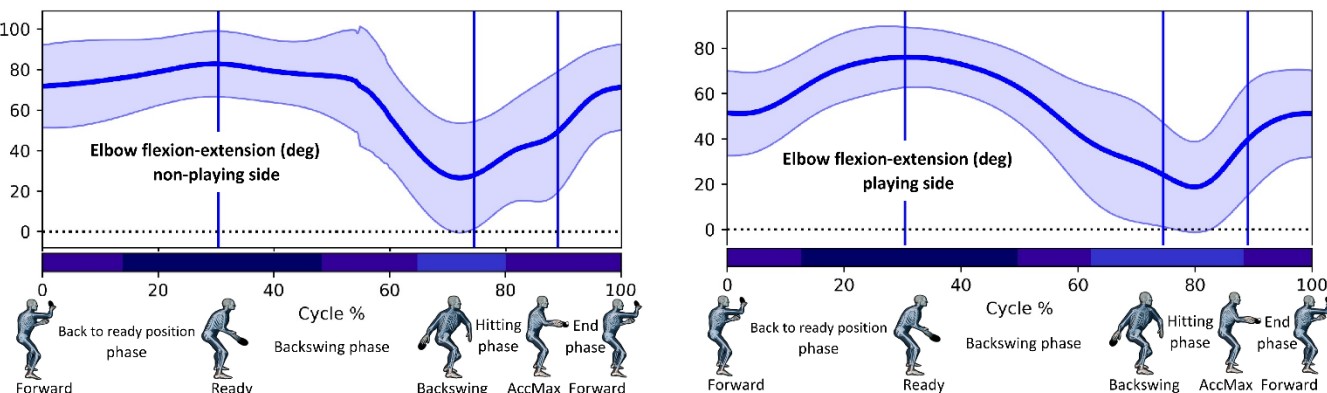

**Figure 4.** Elbow flexion–extension for the non-playing (on the left) and playing side (on the right). The information about variance in the movement exceeding predefined 20, 40, 100, or 150% thresholds was transferred to a color bar representation and placed under each graph. Vertical blue lines indicate the forward, ready, backswing, and AccMax positions in the movement cycle.

Wrist flexion–extension. In the wrist joint, the course of the flexion–extension movement is characterized by a very large SD during the entire cycle, on both sides. The analysis of the average waveform in the non-playing joint shows that, for the most part, it is positioned in a slight lunge, and the movement in this joint is in a very small range (Figure 5). On the playing side, in this joint, at the end of the backswing phase, the players first flex the arm, then straighten until the highest acceleration (AccMax), to about 20 degrees on average, then flex again, to about 10 degrees, and this value of flexion is maintained until 3/4 through the backswing. NFV values are mostly medium in both limbs, in the playing limb in the middle of the swing phase, and around the maximum hand acceleration they are also small.

Wrist supination–pronation: The supination–pronation movement at the wrist joint is again a movement with a large SD during the cycle (Figure 5). On both sides, pronation is noticeable during the hitting phase until the AccMax, with a similar range, but in a different hand position; more pronation on the non-playing side. The non-playing limb at this joint is characterized by a small variation of movement; less than the playing limb.

Wrist radial abduction–adduction: The movement of radial abduction–adduction is characterized by a large SD on both sides. The movement occurs to a very small extent on the non-playing side, and to a slightly larger extent on the playing side (Figure 5). On the playing side, an abduction movement can be observed during the swing phase, with an

adduction movement during the hitting phase. In both limbs a very large (non-playing limb) and extremely large (playing limb) NFV value is noticeable.

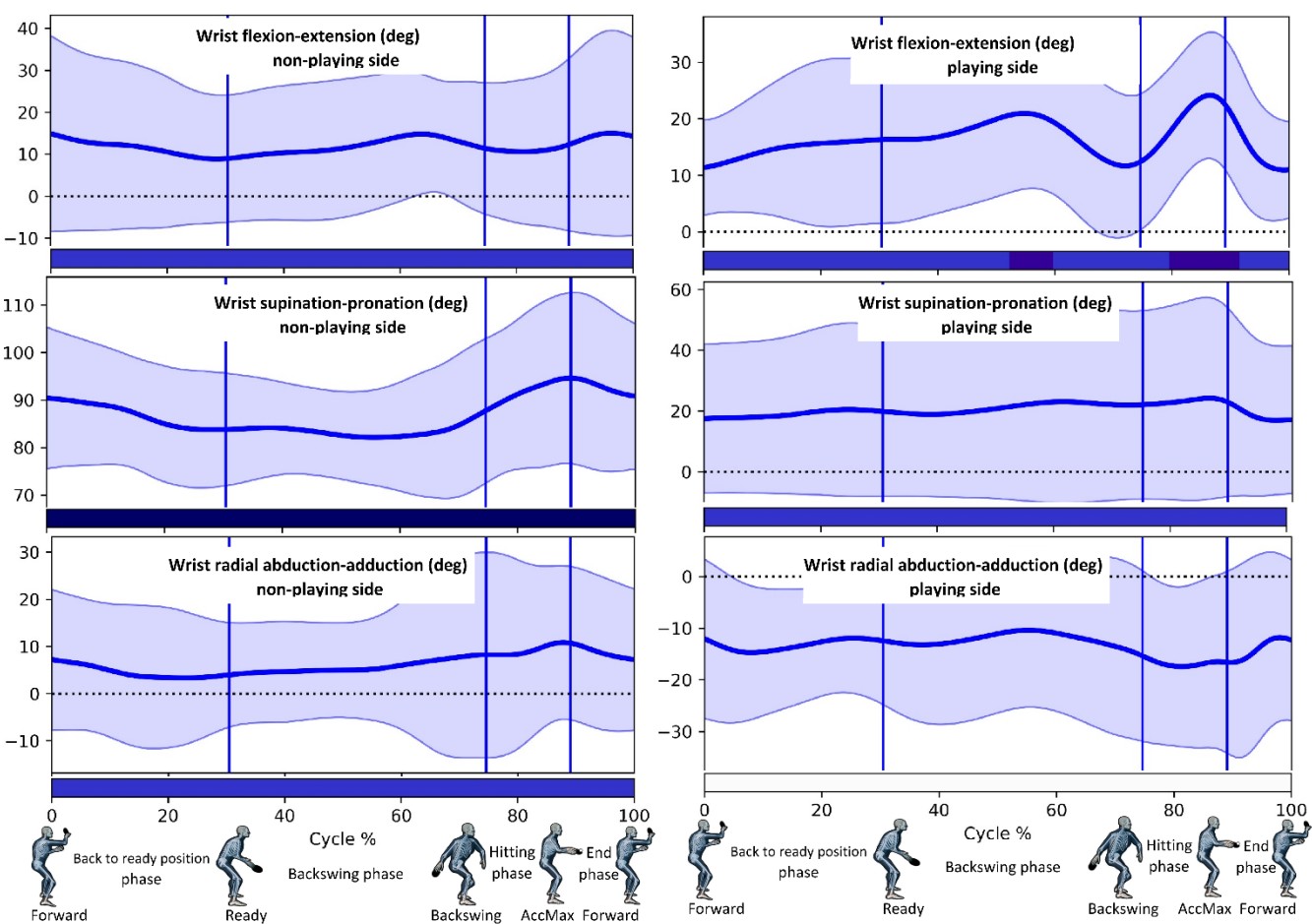

**Figure 5.** Hand kinematics for the non-playing (on the left) and playing side (on the right). Three components of the wrist movement were evaluated: wrist flexion–extension, wrist supination–pronation, and abduction–adduction. The information about variance in the movement exceeding predefined 20, 40, 100, or 150% thresholds was transferred into color bar representations and placed under each graph. Vertical blue lines indicate the forward, ready, backswing, and AccMax positions in the movement cycle.

## 4. Discussion

The purpose of this study was to evaluate the movement pattern in the joints of the non-playing limb during a topspin forehand stroke (played from a ball with backspin) during the whole movement cycle, and to determine the inter-individual variation in this area. The role of the non-playing limb in the performance of the analyzed stroke was determined based on the movement pattern. The present research used a novel method of analysis of variance for repeated measures and classification based on the commonly-used coefficient of variance (CV) for numerical data (NFV).

The analysis of the movements in the joints of the non-playing hand during the topspin forehand stroke indicated that there is a movement about different axes in each joint. It is noteworthy that most of the movements in the joints of the non-playing limb are performed symmetrically to the playing hand, especially in the hitting phase. Of course, the ranges of these movements are different. The timing of key events is also different. Very characteristic is the coordinated movement of flexion and rotation in the shoulder joint and flexion in the elbow joint in the non-playing limb. A rapid change of direction characterizes these movements just before or during the hitting phase, which may indicate the supportive, 'driving' character of these movements. At the elbow joint, first, extension is observed, and

then rapid flexion during the hitting phase (about 60 degrees), the shoulder joint changes from external rotation to internal rotation (about 40 degrees) and from extension to rapid flexion (about 40 degrees). In tennis, Bahamonde [15] found that the non-playing hand acts as a brake to quickly slow down the rotation of the trunk, causing the upper limb to snap forward. The same author pointed out the role of the non-playing limb in decreasing or increasing the torque around different axes of motion throughout the serving cycle in tennis.

The studies available in the literature relating to the non-playing/non-throwing limb have most often focused on throwing competitions or disciplines. Murata [21] stated that the shoulder of a non-playing arm during a baseball pitch must maintain a constant position, while the shoulder of the throwing arm moves in a nearly circular path around it. He suggested that less shoulder joint movement of the non-throwing arm is required to attain a skilled pitch and higher initial ball velocity. Barfield et al. [22] stated that a more extended glove arm elbow and more horizontally abducted glove arm shoulder at the moment of the backswing in the throwing shoulder could prove to be advantageous for performance and possibly be a safer motion for the baseball thrower. In our study, we did not evaluate the movement of the entire shoulder girdle, but from a theoretical point of view, it can be assumed that the shoulder of the non-playing hand, as in throwing disciplines, can provide a stable point around which the contralateral (playing) hand performs the movement; a loop. This hypothesis, evidently, requires further, more thorough research.

In the abduction–adduction movement, symmetrical abduction was found in both shoulder joints during the hitting phase, with an average range of about 180 degrees for the playing hand and about 40 degrees for the non-playing side. The abduction of the arm at the shoulder joint of the non-playing limb has already drawn the attention of researchers in throwing competitions. Fu et al. [23] noted the importance of this movement for throwing technique in children. Coaches should not recommend that children use the non-playing arm for balance or targeting for maximum ball release, as this not only disturbs movement patterns but also neglects the influence of the non-throwing arm on the throwing arm.

Ishida and Hirano [24] observed that the non-throwing arm in the baseball pitching motion contributes to enhancing pitched ball velocity by controlling the moment of inertia of the upper torso and upper extremities along the trunk axis. Perhaps this is also a vital phenomenon of the non-playing limb movements in table tennis. From a theoretical point of view, given the importance of movements about the long axis of the body for the acceleration of the playing hand, the consequences of the adduction of the arm on the non-playing side during the hitting phase should also be investigated. Perhaps, this movement would influence the racket speed, similarly to tennis [15] or when performing triple and quadruple figure skating jumps in figure skating [24]. This could be a manifestation of the principle of conservation of angular momentum. However, in the present study, we could not definitively confirm such use. Most of the participants used the non-playing arm symmetrically during the tasks. It probably helped to keep their balance and maybe stabilized the position of the shoulder girdle. It is also possible that another utilization of the non-playing arm (non-symmetrical, with greater extent of adduction, especially in the hitting phase) would increase the speed of the stroke (conservation of angular momentum) and could be an innovative element in the technique of topspin forehand. The above can be regarded as a practical application of the present study.

The observation and analysis of the distribution and changes in CV values allowed us to observe and indicate variability in the duration of the entire movement cycle in both limbs. This observation confirmed the conclusions of our previous work, in which we found high inter- and intra-individual variability in the kinematics of topspin forehand [5]. However, it was noted that more inter-individual variability (as expressed by scalar CV values) occurs on the non-playing side. The reduced repetition perhaps indicates a greater degree of individualization on the non-playing side, possibly due to less diligence in technique tuition for the non-playing side.

There were some limitations in this study. First of all, as we mentioned earlier, we did not research the kinematics of the whole shoulder girdle. Maybe the results of such research would shed more light on the significance of the non-playing limb during topspin forehand. We also considered only the movement patterns of the limbs, and possibly some other kinematic parameters, such as velocities or accelerations, would give more information. We also have to add that the position of the lower limbs may affect the movements of the upper limbs; therefore, it would be reasonable to conduct research considering this fact in the future.

## 5. Conclusions

Most of the movements in the joints of the non-playing limb are performed symmetrically to the playing one; especially in the hitting phase. Of course, the ranges of these movements are different. This symmetry probably helps to keep the balance and maybe a stable position of the shoulder girdle. A rapid change of direction characterizes these movements in both upper limbs just before or during the hitting phase, which may indicate the supportive, 'driving' character of these movements. The above can be taken as the main practical applications. The observation and analysis of the distribution and changes in CV values allowed us to observe and indicate variability in the duration of the entire movement cycle in both limbs. It was noted that more inter-individual variability occurs on the non-playing side. This phenomenon perhaps indicates a greater degree of individualization on the non-playing side, possibly due to less diligence in technique tuition for the non-playing side. Observing the movement pattern in the non-playing limb during the topspin forehand, we did not find any signs of a use of the principle of conservation of angular momentum to accelerate the racket.

**Supplementary Materials:** The following are available online at https://www.mdpi.com/article/10.3390/sym13112054/s1, Figure S1: Research station, Figure S2: Methodology of graph creation. Left and right shoulder rotation (above graph) as an example of low (**a**) vs. high (**b**) variability results. The variance (bottom graph) information exceeding predefined 20, 40, 100, or 150% thresholds (horizontal blue lines) was transferred to the color bar representation. Vertical blue lines indicate the forward, ready, backswing, and AccMax positions in the movement cycle, Figure S3: Shoulder kinematics for the non-playing (on the left) and playing side (on the right). The movement of the shoulder was evaluated separately for the sagittal plane (flexion–extension), frontal plane (abduction–adduction), and transversal plane (internal–external rotation). The information about variance in the movement exceeding predefined 20, 40, 100, or 150% thresholds was transferred into color bar representation and placed under each graph. Vertical blue lines indicate the forward, ready, backswing, and AccMax positions in the movement cycle, Figure S4: Elbow flexion–extension for the non-playing (on the left) and playing side (on the right). The information about variance in the movement exceeding predefined 20, 40, 100, or 150% thresholds was transferred to a color bar representation and placed under each graph. Vertical blue lines indicate the forward, ready, backswing, and AccMax positions in the movement cycle. Figure S5: Hand kinematics for the non-playing (on the left) and playing side (on the right). Three components of the wrist movement were evaluated: wrist flexion–extension, wrist supination–pronation, and abduction–adduction. The information about variance in the movement exceeding predefined 20, 40, 100, or 150% thresholds was transferred into color bar representations and placed under each graph. Vertical blue lines indicate the forward, ready, backswing, and AccMax positions in the movement cycle.

**Author Contributions:** Conceptualization, I.M.L., Z.B. and S.W.; methodology, I.M.L., Z.B. and S.W.; software, Z.B. and S.W.; validation, I.M.L., Z.B. and S.W.; formal analysis, I.M.L., Z.B. and S.W.; investigation, I.M.L., Z.B. and S.W.; resources, Z.B.; data curation, S.W.; writing—original draft preparation, Z.B.; writing—review and editing, I.M.L., S.W.; visualization, Z.B. and S.W.; supervision, I.M.L. and S.W.; project administration, Z.B.; funding acquisition, Z.B. and S.W. All authors have read and agreed to the published version of the manuscript.

**Funding:** This research received no external funding.

**Institutional Review Board Statement:** The study was conducted according to the guidelines of the Declaration of Helsinki, and approved by the Institutional Ethics Committee of the University School of Physical Education in Wrocław, Poland, IRB number 34/2019.

**Informed Consent Statement:** Informed consent was obtained from all subjects involved in the study.

**Data Availability Statement:** The data presented in this study are available in the file Supplementary Files.

**Conflicts of Interest:** The authors declare no conflict of interest.

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
