# Peer review of "The Role of the Non-Playing Hand during Topspin Forehand in Table Tennis"

_symmetry, doi:10.3390/sym13112054_

Round 1

Reviewer 1 Report

As it is a descriptive study, it would be interesting to extrapolate the conclusions to the field of training. Determine if greater or lesser symmetry would improve technique training. On the other hand, depending on the results and conclusions, a practical application proposal is missing.

Author Response

Dear Reviewers of Symmetry

We appreciate all the constructive comments and valuable observations very much. We also thank you for the effort and time put into the review of our manuscript.

In the following response, each comment has been carefully considered point by point and replied to. Responses to the reviewers are in a dialogue and marked with tab and italics; changes in the revised manuscript are tracked.

Response to Reviewer 2 comments:

As it is a descriptive study, it would be interesting to extrapolate the conclusions to the field of training. Determine if greater or lesser symmetry would improve technique training. On the other hand, depending on the results and conclusions, a practical application proposal is missing.

Thank you very much for this comment. Without prejudging what is better, we added in the Discussion section: “Most of the participants symmetrically used non-playing arm during the tasks. It probably helps to keep the balance and maybe to stabilize the position of the shoulder girdle. It is also possible that other use of non-playing arm (non-symmetrical, with greater extent of ad-duction, especially in the hitting phase) would increase the power of the stroke (the principle of conservation of angular momentum) and could be an innovative element in the technique of topspin forehand. The above can be treated as a practical application of present study”.

Kind regards

Sławomir Winiarski, Ivan Malagoli Lanzoni, Ziemowit Bańkosz

Reviewer 2 Report

This manuscript was designed to evaluate the movement pattern in the joints of the non-playing limb during topspin forehand stroke. The study is easy to follow and well designed. I have some comments before suggest it for publication.  

Abstract Section

The authors could add information with respect to the gap of the study before reporting the aim of the study.

Please add the demographics data of the participants  

Discussion Section

Line 290-296 – I believe it is a methodological aspect of the manuscript, so I would suggest removing it to the Discussion Section.

I would suggest to the authors report how movement in the joints of the non-playing limb during topspin forehand stroke could influence the performance of the athlete.

Do the authors suggest movement patterns in the joints that could improve the performance?

What is the practical implication of the findings of the present study?       

Conclusion Section

The authors should focus on describing the main findings of the study and highlight how the findings will impact table tennis performance. Futures indication is also warranted.

Author Response

Dear Reviewers of Symmetry

We appreciate all the constructive comments and valuable observations very much. We also thank you for the effort and time put into the review of our manuscript.

In the following response, each comment has been carefully considered point by point and replied to. Responses to the reviewers are in a dialogue and marked with tab and italics; changes in the revised manuscript are tracked.

Response to Reviewer 3 comments:

This manuscript was designed to evaluate the movement pattern in the joints of the non-playing limb during topspin forehand stroke. The study is easy to follow and well designed. I have some comments before suggest it for publication.  

Abstract Section

The authors could add information with respect to the gap of the study before reporting the aim of the study.

Thank you very much for this comment. At the beginning of the Abstract, we added: “Theoretical tutorials and scientific literature do not provide information on the proper use of the non-playing hand in table tennis.”

Please add the demographics data of the participants  

We added this information. It stays now: “The study involved 12 male table tennis players (178.7±5.5 cm, 70.0±6.6 kg, 23±3 y) at a competitive level.”

Discussion Section

Line 290-296 – I believe it is a methodological aspect of the manuscript, so I would suggest removing it to the Discussion Section.

We removed all these paragraphs (“The formula for CV = SD / Mean could not be directly applied due to the presence of mean values close to zero. The problem could be resolved thanks to standardized translation towards positive values of the whole range of change (without influencing the amplitude of change). The interpretation of the so-called Normalized Function of Variance (NFV) is similar to CV and differentiates areas of low, average, high, very high or extremely high variability (Figure 2”), because it was described above in the method section.

I would suggest to the authors report how movement in the joints of the non-playing limb during topspin forehand stroke could influence the performance of the athlete.

Thank you very much for this comment. We have already written that “Very characteristic is the coordinated movement of flexion and rotation in the shoulder joint and flexion in the elbow joint in the non-playing limb. A rapid change of direction characterizes these movements just before or during the Hitting phase, which may indicate the supportive, "driving" character of these movements”. We added to the manuscript also a few explanations. It stays now: “Most of the participants during the tasks used non-playing arm in a symmetrical way. It probably helps to keep the balance and maybe stabilize the position of the shoulder girdle.

Do the authors suggest movement patterns in the joints that could improve the performance? What is the practical implication of the findings of the present study?

Thank you for these questions. We indeed suggest, not judging, the other movement pattern. We added this section: “Most of the participants during the tasks used non-playing arm in a symmetrical way. It probably helps to keep the balance and maybe stabilize the position of the shoulder girdle. It is also possible that other use of non-playing arm (non-symmetrical, with greater extent of ad-duction, especially in the hitting phase) would increase the speed of the stroke (the principle of conservation of angular momentum), and could be an innovative element in the technique of topspin forehand. The above can be treated as a practical application of the present study.”

Conclusion Section. The authors should focus on describing the main findings of the study and highlight how the findings will impact table tennis performance. Futures indication is also warranted.

Thank you very much for these comments. We rephrased the Conclusions and added the following: “This symmetry probably helps to keep the balance and maybe stabilize the position of the shoulder girdle. A rapid change of direction characterizes these movements in both upper limbs just before or during the Hitting phase, which may indicate the supportive, "driving" character of these movements. The above can be found as main practical applications.”

Kind regards

Sławomir Winiarski, Ivan Malagoli Lanzoni, Ziemowit Bańkosz

Reviewer 3 Report

This study is concerned with evaluating the progress of motion in the joints of a non-playing limb during a topspin forehand (played after a backspin ball) during a stroke cycle and determining the inter-individual variation in this area.

  • Is there a link between the quality of the non-playing upper limb's movements and the level of the player?
  • The work looks at differences in upper limb movements. But since the position of the lower limbs affects the movements of the upper limbs, I think it would be appropriate to describe that connection, or at least mention it.
  • Do the observed differences in movements of the non-playing upper limb not relate to the position of the lower limbs?
  • Line 131: Was this setting used in previous literature? If so, please cite.

  • Line 346: I recommend to change „our study“ to „this study“ or „the study“.

Author Response

Dear Reviewers of Symmetry

We appreciate all the constructive comments and valuable observations very much. We also thank you for the effort and time put into the review of our manuscript.

In the following response, each comment has been carefully considered point by point and replied to. Responses to the reviewers are in a dialogue and marked with tab and italics; changes in the revised manuscript are tracked.

Response to Reviewer 4 comments:

This study is concerned with evaluating the progress of motion in the joints of a non-playing limb during a topspin forehand (played after a backspin ball) during a stroke cycle and determining the inter-individual variation in this area.

  • Is there a link between the quality of the non-playing upper limb's movements and the level of the player?

Thank you for the above question. Of course, the movement of the non-playing limb may affect the coordination of the whole movement - the stabilization of the position, the coordination of the whole body (or torso), or the movement of the playing limb. We added this explanation in the Introduction.

  • The work looks at differences in upper limb movements. But since the position of the lower limbs affects the movements of the upper limbs, I think it would be appropriate to describe that connection, or at least mention it.

Thank you for this comment. In the Discussion section, we added: “We have to add also that the position of the lower limbs may affect the movements of the upper limbs; therefore it would be reasonable to conduct research considering this fact in the future”.

  • Do the observed differences in movements of the non-playing upper limb not relate to the position of the lower limbs?

Thank you for this question. Probably yes, they do relate, as we mentioned above.

  • Line 131: Was this setting used in previous literature? If so, please cite.

Yes, indeed. We added positions [5, 17, 18] in the main body of the manuscript.

  • Line 346: I recommend to change „our study“ to „this study“ or „the study“.

Thank you, we changed it.

Kind regards

Sławomir Winiarski, Ivan Malagoli Lanzoni, Ziemowit Bańkosz